# Micronutrients in Sepsis and COVID-19: A Narrative Review on What We Have Learned and What We Want to Know in Future Trials

**DOI:** 10.3390/medicina57050419

**Published:** 2021-04-26

**Authors:** Matteo Rossetti, Gennaro Martucci, Christina Starchl, Karin Amrein

**Affiliations:** 1Department of Anesthesia and Intensive Care, IRCCS-ISMETT (Istituto Mediterraneo per i Trapianti e Terapie ad Alta Specializzazione), 90133 Palermo, Italy; mrossetti@ismett.edu (M.R.); gmartucci@ismett.edu (G.M.); 2Division of Endocrinology and Diabetology, Department of Internal Medicine, Medical University of Graz, Auenbrugger Platz 15, 8036 Graz, Austria; christina.starchl@stud.medunigraz.at

**Keywords:** vitamin D, vitamin C, zinc, thiamine, nutrition, critically ill patients, infections, mitochondria, shock

## Abstract

Sepsis remains the leading cause of mortality in hospitalized patients, contributing to 1 in every 2–3 deaths. From a pathophysiological view, in the recent definition, sepsis has been defined as the result of a complex interaction between host response and the infecting organism, resulting in life-threatening organ dysfunction, depending on microcirculatory derangement, cellular hypoxia/dysoxia driven by hypotension and, potentially, death. The high energy expenditure driven by a high metabolic state induced by the host response may rapidly lead to micronutrient depletion. This deficiency can result in alterations in normal energy homeostasis, free radical damage, and immune system derangement. In critically ill patients, micronutrients are still relegated to an ancillary role in the whole treatment, and always put in a second-line place or, frequently, neglected. Only some micronutrients have attracted the attention of a wider audience, and some trials, even large ones, have tested their use, with controversial results. The present review will address this topic, including the recent advancement in the study of vitamin D and protocols based on vitamin C and other micronutrients, to explore an update in the setting of sepsis, gain some new insights applicable to COVID-19 patients, and to contribute to a pathophysiological definition of the potential role of micronutrients that will be helpful in future dedicated trials.

## 1. Introduction

In the USA alone, apart from COVID-19, sepsis affects around 1.5 million people annually [1]. Based on the most recent epidemiological trends, incidence of sepsis is growing [2], with an incidence that is more than 5-fold greater in the elderly population [3]. In a trend analysis conducted from 1993 to 2003, the percentage of severe sepsis cases requiring hospitalization increased from 25% to 44% [4]. In-hospital sepsis mortality has been estimated up to 140% higher compared to annual estimates of mortality due to other causes [5].

Sepsis remains the leading cause of mortality in hospitalized patients, contributing to 1 in every 2–3 deaths [6]. It is the result of a complex interaction between host response and the infecting organism, resulting in life-threatening organ dysfunction, depending on microcirculatory derangement, cellular hypoxia/dysoxia driven by hypotension and, potentially leading to death. All these processes are finely regulated by merging pathways involving a number of cells and mediators.

Standard care for septic patients still involves the first-hour bundle, with the explicit intention of beginning resuscitation and management immediately. Mainstays of treatment are still lactate monitoring, early diagnosis/treatment using cultures and broad-spectrum antibiotics, as well as adequate hemodynamic support to guarantee adequate end-organ perfusion [7]. However, the high energy expenditure driven by a high metabolic state induced by the host response can rapidly lead to micronutrient depletion [8]. This deficiency can result in alterations in normal energy homeostasis, free radical damage, and immune system derangement [6]. In the critically ill septic patient, the adjunctive administration of vitamins and micronutrients, especially in defective scenarios, could lead to a better energy expenditure homeostasis [9]. Moreover, vitamins, and generally micronutrients, despite being neglected for years in the critically ill population, may represent a missing tool in the regulation of processes involved in sepsis, due to their ubiquitous presence and action, the involvement in several biochemical reactions as a cofactor and, in some cases, with indirect genomic and non-genomic effects on the cells involved in the inflammation pathways.

This review will address the topic, including the recent advancement in the study of some micronutrients, including vitamin D, vitamin C, thiamine, and zinc. These are the micronutrients for which, despite controversies, there is some evidence and associations between the disease severity in critically ill patients and their deficiency. In other cases, they have been tested as supplementation in clinical studies.

The field of micronutrients has been entered into clinical studies recently and suffers from methodology biases in clinical studies, since the exploration of such a topic was mainly relegated to pre-clinical interest. However, with its potential for fine-tuning the regulation of biochemical processes and the high evidence of association between disease severity and their deficiency, it is worthy of consideration by clinicians.

In this light, we will give an overview of the actions of micronutrients and their involvement in sepsis and in COVID-19, which has several clinical features in common with severe sepsis.

## 2. Vitamin D

Initially discovered and studied as a major regulator of calcium metabolism, vitamin D also plays an essential role as an immunomodulatory hormone [10] and in several biological activities interfering with the innate and adaptive immune system, with a role even in liver transplant recipients regarding graft function and sepsis incidence [11]. This is also proven by the fact that vitamin D receptors are expressed by immune cells such as lymphocytes, monocytes, macrophages, and dendritic cells [12].

### 2.1. Physiology and Requirements

There are two forms of native vitamin D. Vitamin D_2_ is synthesized from ergosterol and can be found in yeast and sun-dried mushrooms. Vitamin D_3_ is synthesized endogenously from 7-dehydrocholesterol in sun-exposed skin. Both D_2_ and D_3_ are metabolized by CYP2R1 (vitamin D-25 hydroxylase) [12] in the liver to 25-hydroxyvitaminD [25(OH)D], which is further metabolized by CYP27B1 to the active form 1,25-dihydroxyvitaminD [1,25(OH)_2_D] [12], which exerts its endocrine and immune effects by binding to the vitamin D receptor (VDR) in the nucleus [13]. 1,25OHD is usually only needed in advanced renal dysfunction and rare conditions, including hypoparathyroidism.

The main site of conversion of 25(OH)D is the kidneys. Evidence shows that circulating levels of 25(OH)D maintained in the range of 40–60 ng/mL are associated with the lowest risk of several types of cancer, and cardiovascular and autoimmune diseases [14]. In order to maintain the blood levels in the range of 20–40 ng/mL, with minimal sun exposure, an adult would require the ingestion of 4000–6000 IU daily [15]; however, daily intakes using standard enteral/parenteral nutrition formulas rarely exceed 500 IU daily.

### 2.2. Vitamin D and Immunity

Vitamin D has a plausible link with response to infection. Macrophages and monocytes express CYP27B1 as a response to cytokines and IFN-γ. This enzyme converts 25(OH)D in the active form 1,25(OH)2D25, which is able to enhance macrophage and monocyte activity by the stimulation of the production of cathelicidin (LL-37), which acts by destabilizing microbial membranes [16]. Furthermore, it exerts antiviral effects by disrupting viral envelopes and altering the viability of host target cells.

In a mouse model, Horiuchi et al. found a low expression of the inflammatory molecule iTXB_2_ in mice receiving oral 1,25(OH)_2_D and intraperitoneal LPS compared to controls who were not receiving the vitamin D metabolite. A significant reduction in mortality was noticed [17].

As known, this modulation process is widely seen in clinical scenarios such as sarcoidosis and tuberculosis, explaining why, for example, some patients affected with granulomatous disorders develop hypercalcemia and hypercalciuria [18]. The upregulation of CYP27B1 also plays a role in regulating lymphocyte activity (reduces Th1 and Th17 activity and stimulates Th2 and Treg). Moreover, 1,25(OH)_2_D modulates tolerance in antigen-presenting cells (APC) by decreasing the expression of major histocompatibility complex class II (MHC-II) [19]. This leads to a decrease in IL-12 production and an increase in IL-10, with tolerogenic effects [20].

Even endothelial function is influenced by vitamin D. Several experimental studies have shown that it can modulate vascular permeability via multiple genomic and extra-genomic pathways. For example, 1,25(OH)_2_D is a transcriptional factor for endothelial nitric oxide synthase (eNOS), able to cause an upregulation of the gene expression augmenting nitric oxide production [21].

This is a potential role that may be interesting for the prevention and treatment of patients with severe cases of COVID-19, given that microangiopathy, coagulopathy, and thrombosis are frequent in COVID-19, and vitamin D deficiency is associated with a prothrombophilic profile, potentially reversible with vitamin D supplementation [22]. In fact, high dose vitamin D supplementation has been associated with reduced in vitro thrombin generation and decreased clot density.

These effects use non-genomic pathways including adenylyl cyclase/cyclic adenosine monophosphate (AC/cAMP) and inositol triphosphate/diacilglycerole (IP_3_/DAG), which lead to an augmentation of intracellular calcium concentration [23].

Multiple studies have reported vitamin D effects also on gut integrity and intestinal homeostasis, showing an ability to alleviate intestinal damage from bacterial lipopolysaccaride [24]. Moreover, vitamin D can increase the expression of epithelial membrane junction proteins, crucial when facing bacterial translocation events.

Vitamin D’s role in modulating adaptive immunity was originally observed on clonal human T-cell-expressing VDR [25]. It seems that resting T cells do not express VDR, while peripheral T cells do, making them a target of 1,25(OH)_2_D produced by macrophages and monocytes involved in the inflammatory response [25]. Vitamin D promotes a shift from Th1 and Th17 to Th2 and Treg immunity by enhancing Th2 cytokine expression while inhibiting Th1. This leads to the suppression of an uncontested proinflammatory state [25,26], even playing a potential role in protection from autoimmune diseases. This role in modulating inflammation is also evident in vitamin-D-deficient individuals, where CD4/CD8 ratios decrease as an indicator of immune activation [27], while the administration of 5000–10,000 IUs of D_3_ can increase CD4/CD8 ratio [28,29].

### 2.3. Vitamin D in the Critically Ill: The Septic Patient

In a large study involving more than 3000 critically ill patients, vitamin D deficiency was a significant predictor of sepsis and carried a 1.6-fold increase in mortality [30]. Several observational studies have reported a connection between low levels of 25(OH)D and the incidence of sepsis; data also support the link between low serum vitamin D levels and the increase in morbidity and mortality in septic, critically ill patients. The reasons seem to be related to the effects of 1,25(OH)2D on the expression of pro-inflammatory cytokines of T_H_1(IL-2, IFN-γ, TNF-α) and T_H_17 (IL-17, IL-12) [31,32,33].The role of vascular reactivity is under debate: lower levels of vitamin D_3_ are associated with worse outcomes, but vitamin D may, at the same time, exert non-genomic actions on endothelial cells to prevent extravascular leakage, and it may be reduced in its plasmatic levels by the vascular leakage itself due to systemic inflammation [34].Vitamin D’s effects seem to encompass not only the modulation of the proinflammatory status, but also the local pathogen’s control: Youssef and colleagues showed how the concentration of 50,000–90,000 IU/mL of D_3_ was able to inhibit the growth of or even kill strains of *Staphylococcus aureus*, *Klebsiella pneumoniae*, *Escherichia coli*, and *Streptococcus pyogenes* [35]. As a direct antimicrobial role becomes better understood, especially considering the modulating effect exerted by 1,25(OH)_2_D after LPS exposure, strong evidence connects vitamin D metabolites to a decrease in pro-inflammatory status, e.g., in yeast-induced sepsis [36]. In addition to basic biological research, some observational studies have explored vitamin D’s role in the clinical setting. One observational study pointed to a connection between vitamin D plasma concentrations and respiratory infection [37], where Ginde et al. observed an inverse relationship between 25(OH)D levels and the incidence of upper respiratory infections (URI), data corroborated by Sabetta and colleagues’ study, in which 25(OH)D levels greater than 38 ng/mL were associated with a 2-fold decrease in URI incidence [38]. In the critically ill population, several studies have revealed a high prevalence of poor vitamin D status [39]: in a single-center study, the prevalence of 25(OH)D < 24 ng/mL was 79% [40], though lacking any association with mortality or hospital-acquired infections. In contrast, a retrospective study of 437 ICU patients showed a significant correlation between low vitamin D levels and 25(OH)D < 20 ng/mL and mortality [41]. In a study of 70 patients divided into three groups, Jeng et al. found vitamin D insufficiency in 100% of critically ill patients admitted with sepsis in the ICU (group 1) and in 92% when considering the non-septic ICU group (group 2), compared with 66.5% in the control group of normal healthy individuals (group 3) [42]. In a case-control cohort study of 36 ventilated patients admitted to the ICU, the group receiving a high-dose intramuscular injection of vitamin D obtained a significant reduction of ventilation days and length of stay.

A summary of the potential positive effects of vitamin D in sepsis is presented in Figure 1.

## 3. Vitamin C

### 3.1. Physiology and Requirements

Involved in several biosynthetic and metabolic processes, vitamin C is essential for collagen and carnitine [43], and neurotransmitter synthesis [44] plays an antioxidant role [45], acting as an immunomodulatory agent [46] (Figure 2). The level considered normal in plasma [47], about 50 µmol/L, according to the EU food safety authority, can be achieved by an intake of 90 mg/day for men and 80 mg/day for women. This is the plausible solution, at a population level, to avoid scurvy, but it has been not demonstrated that it is a sufficient intake in case of viral infections of other processes with a high level of antioxidant consumptions. The overt vitamin C deficiency can be diagnosed by a plasma level below 11 µmol/L, but it is rarely checked in hospitalized patients, and even among the most severe patients, this feature is definitely neglected [47]. This happens despite the fact that we know that the level of vitamins decreases rapidly in sepsis, trauma, surgery, and, recently, in COVID-19 patients.

In septic patients, vitamin C is involved in the modulation of the proinflammatory and procoagulant state believed to induce vascular-ischemic induced multiple organ injury [48]. In addition, vitamin C seems to reduce platelet aggregation by modulating surface P-selectin expression [49], attenuate hypothalamic neuronal damage, and prevent immunosuppression, in addition to inducing endogenous vasopressor synthesis [50]. As with studies on vitamin D, several studies have found a reduction in vitamin-C-circulating levels in septic patients admitted to the ICU, and deficiency may be exacerbated by the reduction in cell uptake due to inflammatory cascade activation (TNF-α and IL-1β can down-regulate the ascorbate-specific transporter [51]). On the other hand, plasma concentration can lower (<10 micromol/L) in the first 24 h after septic onset, an event that is strongly associated with an increase in the severity of organ dysfunction and mortality.

### 3.2. Vitamin C in the Critically Ill Septic Patient

Within the ICU population, to achieve normal plasma concentration and counteract organ dysfunction, high dose administration is necessary (3 g/daily) for 72 h. This may reduce vasopressor requirements in septic shock and mortality in the ICU septic population [52], though more evidence will be needed. This hypothesis has a strong pathophysiological plausibility and relies mainly on a small and controversial before-after retrospective study [53]. Several RCTs were unable to demonstrate a reduction in mortality through the use of vitamin C, potentially due to several limitations, such as heterogeneous populations or too severe patients and a lack of early administration [54]. The principal criticism is the use of such a therapy in cases of advanced severe septic shock, at least in light of mortality as an outcome, since, as demonstrated for vitamin D, the action of micronutrients on severely ill patients may be less relevant, likely because the severity of the organ failure is the result of several metabolic pathways that cannot easily be improved upon.

Another relevant topic on vitamin C administration is related to the pharmacokinetic aspect. In fact, being a water-soluble vitamin, it is rapidly excreted if not used. For this reason, due to its rapid use in the oxidative process, the main results were reached with repeated administration every 6 h. In the largest trial of intravenous vitamin C in sepsis-associated ARDS, the CITRIS-ALI trial [55], patients were given placebo or vitamin C at a dose of 50 mg/kg every 6 h for 4 days. This means about an average dose of 3.5 g every 6 h in adults. Looking at the stated primary study outcomes, vitamin C did not improve markers of inflammation, vascular injury, or organ dysfunction. However, there were statistically significant benefits in three clinically relevant outcomes: mortality (*p* = 0.03), duration of ICU-free days (*p* = 0.03), and hospital-free days (*p* = 0.04). As a matter of fact, examining the data, during the 4-day vitamin C administration, mortality was 81% lower in the vitamin C group, but after the cessation of study drug administration, there was no difference between the two trial groups. This study, as well as other similar negative studies, poses a relevant question about seeking proper evidence in critically ill patients, when the research objectives just apparently, contrast with the clinical effects.

## 4. Other Micronutrients

### 4.1. Thiamine (Vitamin B1)

Thiamine is a cofactor for several enzymes involved in aerobic carbohydrate metabolism, maintenance of cellular redox homeostasis, and synthesis of adenosine triphosphate [56] (Figure 2). In particular, thiamine is needed to convert pyruvate into acetyl-CoA, allowing entry into the citric acid cycle and aerobic metabolism. The human body has limited storage abilities within skeletal muscle, heart, kidney, and brain [57], and due to its quick turnover, without supplementation, deficiency can develop in just two weeks, with a clinical spectrum ranging from cardiac beriberi to Wernicke’s encephalopathy [58].

In septic patients, thiamine deficiency is commonly found, with a prevalence of 20% to 71%: 20% of septic patients and 71% of those presenting with septic shock exhibit thiamine deficiency (<9 nmol/L) [59,60] (normal range of value is considered to be within 33–99 ng/mL). Several mechanisms have been identified to explain the association between thiamine deficiency and sepsis, though it remains unclear whether the deficiency can contribute as a cause of sepsis or if it is just a consequence. What is clear is that by decreasing pyruvate dehydrogenase activity (needed to convert pyruvate in acetyl-CoA to enter the citric acid cycle), thiamine deficiency can increase anaerobic metabolism and lactic acid production, possibly worsening sepsis-related consumption of endogenous antioxidants, a hallmark of septic multi-organ damage [61]. Moreover, its antioxidant activity is manifested through the prevention of lipid peroxidation and oleic acid oxidation. Therefore, in De Andrade and colleagues’ murine model, thiamine deficiency was associated with oxidative stress and a proinflammatory state [62]. The clinical consequence is, however, unclear: in a large randomized clinical study, the administration of thiamine in ICU patients considered to be thiamine deficient did not improve mortality or ICU stay, but was associated with a lower rate in progression to renal replacement therapy [63,64]. In a small observational study, Marik et al. suggested that the combination of hydrocortisone (50 mg every 6 h for 4 days), vitamin C (1.5 mg every 6 h for 4 days), and thiamine (200 mg every 12 h for 4 days) significantly improved outcomes in patients with sepsis and septic shock [65].

### 4.2. Zinc

Zinc homeostasis may be fundamental in the organism’s reaction to sepsis. As an essential trace element, it works as a co-factor for several enzymes and its deficiency leads to delayed wound healing, lymphopenia, and a high incidence of infection [66]. Concentrating on the immune system, zinc is crucial for T-cell maturation and differentiation [67] and protects against the premature apoptosis of immature T cells, which can lead to altered Th1/Th2 ratios and, eventually, to total T-cell count decrease [68]. On the cellular level, zinc serves as a second messenger and is involved in the development of pro-inflammatory cytokines by monocytes [69] presentation, of major histocompatibility complex type II by dendritic cells [70], and proliferation of T cells [71] via IL-2 stimulation. As part of the acute phase reaction in sepsis, zinc deficiency is linked to an increase in TNF-α and IL-6, and to explain this phenomenon, some authors have proposed a model of redistribution of zinc mediated by cytokines [72], and a reduced concentration of serum zinc has been found in septic patients admitted to ICUs for no alimentary reason [73] (Figure 2). A series of studies has found that exposure to LPS and pro-inflammatory cytokines such as IL-6 led to an upregulation of the protein ZIP14 in the liver [74], where it serves as a zinc transporter and is essential for the phosphorylation of c-Met during liver regeneration [75]. In a murine model, ZIP14 ko mice exposed to LPS did not show hypozincemia, but developed hypoglycemia as a mark of hepatic glycemic dysregulation [75]. Where the ZIP14 protein can be upregulated, by contrast, hypozincemia begins within 9 hours [76], and the redistribution of zinc in the liver has been associated with lower degrees of accumulation of superoxide anion and necrotic cell death in the organ [77], suggesting a possible protective role of zinc in acute phase liver dysfunction. On the other hand, a decrease in serum zinc concentration can lead to a downregulation of lymphopoiesis and an upregulation of myelopoiesis, showing a sort of reprogramming of the immune response with a shift from adaptive-based to innately-predominant during hypozincemia [78]. Despite the unclear, but potential, physiological role of zinc redistribution, its serum reduction could lead to higher levels of proinflammatory cytokines, higher oxidative stress, lipid peroxidation, and damage to DNA [79]. Though zinc’s role is largely an unexplored path, several studies have shown a correlation between low serum zinc concentration and higher SOFA scores [80], and sepsis non-survivors had much lower zinc concentrations than patients with favorable outcomes [81].

Data regarding a potential beneficial role of zinc supplementation in septic patients still fail to reach statistical significance, though a possible role might be played by albumin, which is the main zinc serum transporter and one of the most important negative acute phase proteins [82,83]. For this reason, more evidence is needed to implement zinc administration in standard sepsis treatment and care, even as a possible biomarker in terms of morbidity and outcome.

## 5. COVID-19 and Micronutrients: What Is Known

SARS-CoV-2 infection resulting in COVID-19 has reached an unexpected, worldwide burden in terms of morbidity and mortality, with 5% of patients hospitalized among all those who tested positive and 20% of those hospitalized developing a severe illness [84]. The most common clinical presentation includes fever (70–90%), dry cough (60–86%), shortness of breath (53–80%), fatigue (38%), myalgias (15–44%), nausea/vomiting or diarrhea (15–39%), headache, weakness (25%), and rhinorrhea (7%). In some cases, anosmia/ageusia can be the presenting symptom (3%). Common laboratory findings include lymphopenia (83%), elevated inflammatory markers like erythrocyte sedimentation rate (ESR), C-reactive protein (CRP), ferritin, IL-1, and IL-6. Chest X-rays often reveal bilateral infiltrates with ground glass opacities [85]. A study of 20,133 hospitalized patients in the UK found that 17.1% had been admitted to high-dependency units or ICUs [86], prompting an exhausting effort by the health system to counteract the pandemic. Impaired function of the heart, brain, liver, lung, kidney, and coagulation systems have been observed, so that approximately 17–35% of hospitalized patients are currently treated in the ICU, due to hypoxemic respiratory failure in the most common scenario. Interestingly, in the case of COVID-19, the clinical picture of the severe cases requiring ICU admission is characterized by multi-organ failure, with many tracts very similar to those of severe sepsis. Therefore, the mechanisms of disease also seem to have some similarity since in COVID-19 as well in severe sepsis and septic shock, the cause of multi-organ failure is not due to a “cytopathic” effect of the bacteria or the virus, but mainly due to the host’s response to the infection.

COVID-19 therapy, especially when treating ICU patients, is strictly supportive, including mechanical ventilation, extracorporeal life support systems such as veno-venous ECMO and antibiotic therapy in the case of bacterial over-infection. Non-specific anti-viral therapy has been proven to be effective in ICU patients. There might be a role, though, for micronutrient supplementation in deficient patients. In the current scenario of limited health resources, it would be important to adopt any adjuvant treatment that may contribute to a better outcome if it is inexpensive and with few or unimportant side effects at tested doses.

### 5.1. Vitamin D and COVID-19

As already described above, vitamin D as an immunomodulatory agent has a strong rationale also in COVID-19. The risk of developing respiratory tract infections is reduced two-fold in adults with a higher serum concentration of 25(OH)D (>38 ng/mL) [38], and the role of 1,25(OH)_2_D in exerting anti-viral activity and modulating immune response by stimulating cathelicidin release is well known. This leads to the suppression of proinflammatory cytokine release [87]. Furthermore, 1,25(OH)_2_D specifically acts as a modulator of the renin-angiotensin pathway and is able to downregulate angiotensin-converting enzyme-2 expression, which is known to be the entry receptor for SARS-CoV-2 in cells [88]. Recently, the effect of a single dose of 200,000 IU of vitamin D_3_ on hospital length of stay in patients with COVID-19 was tested, showing no effect between the vitamin D_3_ and the placebo group for the primary or secondary end points [89]. This study is paradigmatic of how the basic science should be deeply known to start a clinical trial on the topic to avoid the risk of eventually misleading negative conclusions [90]. In fact, though a loading dose is imperative in acute settings to improve vitamin D levels rapidly, it is unphysiological to give only a loading dose not followed by a maintenance dose [91]. Despite the practical advantage, a single or annual dose has repeatedly been shown to be ineffective or even harmful for respiratory tract infections and musculoskeletal outcomes. Considering the population enrolled in the study, only 115 of 240 patients were vitamin D deficient at baseline (25OHD < 20 ng/mL), with no information on the proportion of patients with severe deficiency (25OHD < 12 ng/mL). Finally, symptom onset was 10 days before the intervention, so the infection likely took place well over two weeks before the intervention. This topic of the right intervention time returns in many studies approaching micronutrients because it is quite impossible that a single intervention can be the only reason for a change in prognosis when a wide intersection of different pathways has been started with superimposing circles.

### 5.2. Vitamin C and COVID-19

Vitamin C could exert many potentially beneficial roles in counteracting SARS-CoV-2 infection: antiviral, immunomodulatory, anti-inflammatory, and antioxidant effects coexist in molecular pharmacodynamics. In vitro studies have confirmed that vitamin C alone is able to suppress the replication of some viral species, such as herpes simplex-1, influenza A, polyvirus type 1, and rhynovirus [92]. In vivo, vitamin C supplementation can reduce the incidence of postherpetic neuralgi [93] and the duration and severity of the common cold [94]. High-dose vitamin C treatment can also reduce symptoms in patients affected with acquired immune deficiency syndrome (AIDS), being able to ameliorate even the severity of opportunistic infections [95]. Vitamin C is also able to modulate the release of proinflammatory cytokines, and in mice models led to augmented release of interferon, thus being able to reduce lung inflammation in viral pneumonitis [96]. With regard to COVID-19, the combination of vitamin C and quercetin has shown promising synergic antiviral activity [97] and can lead to augmented endothelial repair in widespread microvascular and microvascular thrombosis with increased permeability [98]. In a Chinese trial, high IV dose vitamin C (10 g/day for moderate cases and 20 g/day for severe cases for 7–10 days) was able to shorten the hospital stay by 3–5 days in 50 patients [99]. In another randomized controlled pilot-trial in three hospitals in China on 56 critically ill COVID-19 patients, 24 g of vitamin C was not able to improve the primary outcome (invasive mechanical ventilation-free days in 28 days) and the 28-day mortality (*p* = 0.27), but it was able to improve the PaO_2_/FiO_2_ ratio in the treatment group on day 7 (229 vs. 151 mmHg, 95% CI 33–122; *p* value = 0.01) as well as reduce the value of IL-6 in the treatment group on day 7 (*p* = 0.04) [100]. Both of the reached positive outcomes were clinically relevant and should prompt further investigations on the topic.

## 6. Conclusions

Micronutrients contribute greatly to the human body’s homeostasis and metabolism. For decades, they have been considered an ancillary concern in critically ill patients. However, with the current need for a new increase in survival for critically ill patients, they should enter any clinical consideration in daily practice. As another side of the coin, research on this topic should consider not only mortality, since in severely ill patients the outcome is too often confounded by concomitant factors; but reliable and clinically sensitive surrogate outcomes, including the functional recovery of daily activities, should be explored in the coming years. 

## Figures and Tables

**Figure 1 medicina-57-00419-f001:**
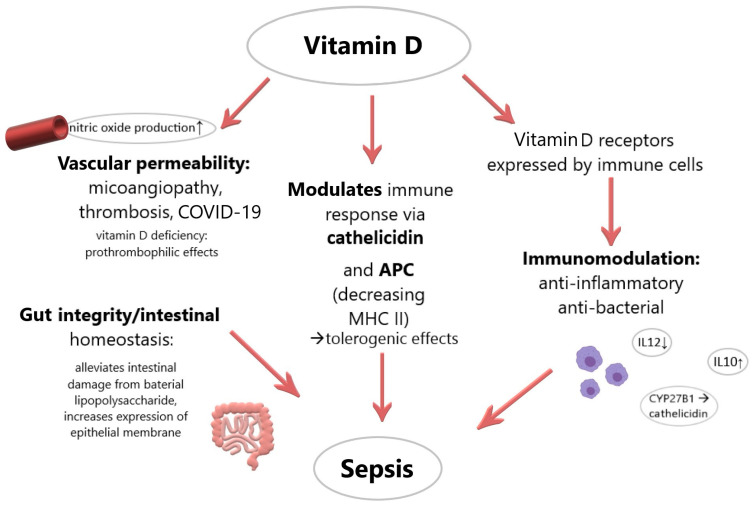
Summary of potential implications of vitamin D during sepsis. (APC: antigen-presenting cell; MCH II: major histocompatibility complex, class II; IL-12: interleukin 12; IL-10: interleukin 10).

**Figure 2 medicina-57-00419-f002:**
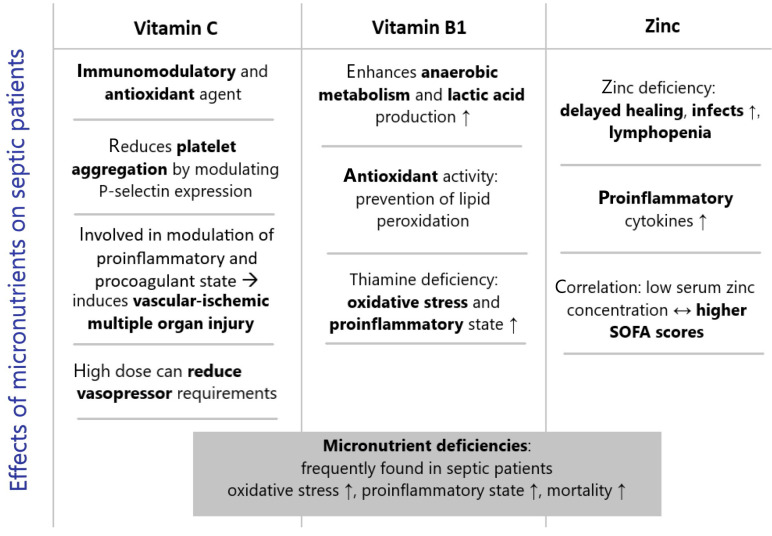
Effects of micronutrients on septic patients, other than vitamin D.

## Data Availability

Not applicable.

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
