# Peer review of "Micronutrients in Sepsis and COVID-19: A Narrative Review on What We Have Learned and What We Want to Know in Future Trials"

_medicina, 2021, doi:10.3390/medicina57050419_

Round 1

Reviewer 1 Report

authors)

STRUCTURE

  • The manuscript is properly structured.
  • It would be advisable to include a section on methodology.

TITLE AND ABSTRACT

  • The abstract should inform that the type of study.
  • Include in the abstract a brief summary of narrative review and implications for future research.

INTRODUCTION

  • Lines 42 and 43: Add reference
  • Line 48: Add reference
  • Background is scarce. More explanation and approach to the topic is needed in the introduction. Further elaboration on sepsis, its relation to COVID-19 and current knowledge on the use of micronutrients.
  • Why have vitamins D, C, thiamine and zinc been specifically investigated? For which nutrients is there evidence? What is the evidence?
  • Line 70: what are these common clinical features? go deeper.
  • Specify the key questions identified for the review topic.
  • The hypotheses is missing.
  • Do not use the first person in scientific publications. Applicable to the entire document “we will...”
  • Highlight what this review adds to the scientific evidence.

MATERIAL AND METHODS

  • A methodology section has not been included. This aspect is very important to assess the quality of the review and to check its scientific scope.
  • It is necessary at least to indicate the dates on which the search was carried out, in which databases as well as the inclusion and exclusion criteria of the articles reviewed.
  • Description of the literature search is missing.
  • Specify the process for identifying the literature search (eg, years considered, language, publication status, study design, and databases of coverage).

RESULTS

  • Line 81 and 82: add reference
  • Line 93: add reference
  • Line 123: add reference
  • Line 130: add reference
  • Line 146: what does "*" mean in the text?
  • Line 190: add reference
  • Line 195: add reference
  • Line 234 and 236: add reference
  • Line 313: add reference
  • Line 326: add reference
  • Line 340: add reference
  • Information on thiamine and zinc in patients with COVID-19 is missing.

DISCUSSION

  • Discuss: 1) research reviewed including fundamental or key findings, 2) limitations and/or quality of research reviewed, and 3) need for future research.
  • Provide an overall interpretation of the narrative review in the context of clinical practice.

REFERENCES

  • References do not follow the indicated style. Review.

Author Response

STRUCTURE

  • The manuscript is properly structured.
  • It would be advisable to include a section on methodology.

A: We believe this is not applicable in the context of a narrative review. 

TITLE AND ABSTRACT

  • The abstract should inform that the type of study.
  • Include in the abstract a brief summary of narrative review and implications for future research.

A: Thank you, we improved the abstract following your comments

INTRODUCTION

  • Lines 42 and 43: Add reference
  • Line 48: Add reference
  • Background is scarce. More explanation and approach to the topic is needed in the introduction. Further elaboration on sepsis, its relation to COVID-19 and current knowledge on the use of micronutrients.
  • Why have vitamins D, C, thiamine and zinc been specifically investigated? For which nutrients is there evidence? What is the evidence?
  • Line 70: what are these common clinical features? go deeper.
  • Specify the key questions identified for the review topic.
  • The hypotheses is missing.
  • Do not use the first person in scientific publications. Applicable to the entire document “we will...”
  • Highlight what this review adds to the scientific evidence.

A: We generally improved the introduction following your suggestions.

MATERIAL AND METHODS

  • A methodology section has not been included. This aspect is very important to assess the quality of the review and to check its scientific scope.
  • It is necessary at least to indicate the dates on which the search was carried out, in which databases as well as the inclusion and exclusion criteria of the articles reviewed.
  • Description of the literature search is missing.
  • Specify the process for identifying the literature search (eg, years considered, language, publication status, study design, and databases of coverage).

A: We respectfully disagree about methodology and believe t is not applicable in the context of a narrative review. 

RESULTS

  • Line 81 and 82: add reference
  • Line 93: add reference
  • Line 123: add reference
  • Line 130: add reference
  • Line 146: what does "*" mean in the text?
  • Line 190: add reference
  • Line 195: add reference
  • Line 234 and 236: add reference
  • Line 313: add reference
  • Line 326: add reference
  • Line 340: add reference
  • Information on thiamine and zinc in patients with COVID-19 is missing.

A: An update on references and their use has been done, also with a new and more recent search

DISCUSSION

  • Discuss: 1) research reviewed including fundamental or key findings, 2) limitations and/or quality of research reviewed, and 3) need for future research.

A: we included a paragraph on limitation and need for future research

  • Provide an overall interpretation of the narrative review in the context of clinical practice.

A: We extended and improved the conclusions paragraph

Reviewer 2 Report

I accepted it 

Author Response

Thank you for accepting this manuscript.

Round 2

Reviewer 1 Report

No further comments 

This manuscript is a resubmission of an earlier submission. The following is a list of the peer review reports and author responses from that submission.

Round 1

Reviewer 1 Report

STRUCTURE

The structure is appropriate for a review, but it would have been more interesting do a systematic review.

TITLE AND ABSTRACT

The title should inform that the type of study.

INTRODUCTION

  • Line 50: add references
  • Explain, in more depth, the rationale for the review in the context and the applicability of the review.
  • State specific objectives.
  • Why specifically vitamin D and vitamin C?

MATERIAL AND METHODS

  • It would be very useful if the materials and methods had been described in order to assess the methodological quality of the study, as well as the quality of the studies that have been included in this review.
  • It would be important to know which databases the articles come from and whether any filters have been applied.

VITAMINA D

  • Line 67: add references
  • Line 70: which effects?
  • Line 79: provide the reference of these 500 IU per day.
  • Do not use the first person in scientific publications. Applicable to the entire document (Lines 91…)
  • On several occasions studies or scientific statements are named that are not referenced, add the references. Applicable to the whole document (Lines 101, 106, 109, 132…)
  • The scientific evidence should be presented or discussed and provide the authors' own conclusion on the use of vitamin D in critically ill patients and its dosage. This conclusion is not found in the section.
  • Figure 1 is not named in the section.
  • Figure 1 is self-made? If so, please indicate.
  • Clarify abbreviations: APC, MHC, etc. on the bottom of the Figure 1.

VITAMINA C

  • Describe the same sections as for vitamin D
  • Figure 2 is not named in the section.
  • Figure 2 is self-made? If so, please indicate.
  • If other micronutrients have not yet been discussed, do not include them in figure 2, vitamin C section.

OTHER MICRONUTRIENTS

  • Describe the same sections as for vitamin D
  • There are no other important micronutrients besides thiamine and zinc? explain why.

COVID-19 AND MICRONUTRIENTS

  • On several occasions studies or scientific statements are named that are not referenced, add the references. Applicable to the whole document (Lines 284, 288, 292…)

REFERENCES

  • References do not follow the style indicated.
  • Various types of references are mixed.
  • There are 110 bibliographic references and most of them are more than 10 years old.

Reviewer 2 Report

This review is a nice reminder that vitamins exist in evolution for vital reasons  and could be playing an important role in regulating the inflammation  of sepsis if only it could be found. However,  sepsis itself remains a mystery at the molecular level. Until  chemical alignments are  better informed it’s unlikely that vitamins  will be able to fit into any niche in sepsis except empirically.

This well written studies stresses pathophysiology  because there is no real chemical molecular biology that fits the premise. The role of vitamins in treating sepsis remains in a morass and is yet  to attain evidence-based medicine. Indeed the molecular biology of sepsis is unknown and since this is the case how can we fit vitamins into unknown molecular architecture.

However, It is important to remind us of this important subject without going into too much detail and providing references to search the field.

I  recommend taking out COVID-19 because it just recites what it is and really has nothing to do with making progress and vitamins. Indeed the problem with vitamins is vitamins which as cofactors require detailed biochemistry in a targeted way.

I suggest a table by that identified to 3 or 4 high priority and specific unanswered questions and how to answer them in ways that elevate  the field out of its doldrums.

in summary, this is a well written, reasonably constructed, and timely  reminder  of the potential importance of vitamins in sepsis and gives some evidence of why they may or may not work. However evidence-based science is not available

can take the field forward his answer